# Complementary Transcriptomic and Proteomic Analysis Reveals a Complex Network Regulating Pollen Abortion in GMS (*msc-1*) Pepper (*Capsicum annuum* L.)

**DOI:** 10.3390/ijms20071789

**Published:** 2019-04-11

**Authors:** Qing Cheng, Ting Li, Yixin Ai, Qiaohua Lu, Yihao Wang, Liang Sun, Huolin Shen

**Affiliations:** 1Beijing Key Laboratory of Growth and Developmental Regulation for Protected Vegetable Crops, China Agricultural University, Beijing 100193, China; chengqing2013@126.com (Q.C.); 15073196720@163.com (T.L.); aiyixin0129@126.com (Y.A.); lqh12261842@163.com (Q.L.); yhwang0906@126.com (Y.W.); liang_sun@cau.edu.cn (L.S.); 2Department of Vegetable Science, College of Horticulture, China Agricultural University, No. 2 Yuanmingyuan Xi Lu, Haidian District, Beijing 100193, China

**Keywords:** pepper, proteomic analysis, transcriptomic analysis, genic male sterility, tapetum

## Abstract

Pepper (*Capsicum annuum* L.) is a globally important horticultural crop. Use of the genic male-sterile (GMS) line enables efficient commercial hybrid pepper seed production. However, the mechanisms of pepper GMS functioning remain unclear. In this study, we used proteomic and transcriptomic analysis to identify proteins and genes related to genic male sterility. A total of 764 differentially expressed proteins (DEPs) and 1069 differentially expressed genes (DEGs) were identified in the proteomic and transcriptomic level respectively, and 52 genes (hereafter “cor-DEGs-DEPs” genes) were detected at both levels. Gene ontology (GO) and Kyoto Encyclopedia of Genes and Genomes (KEGG) analysis identified 13 DEPs and 14 DEGs involved in tapetum and pollen development. Among the 13 DEPs identified, eight were involved in pollen exine formation, and they were all up-regulated in the fertile line 16C1369B. For the 14 DEGs identified, *ABORTED MICROSPORES* (*AMS*) and *DEFECTIVE IN TAPETAL DEVELOPMENT AND FUNCTION1* (*TDF1*) were involved in tapetum development, and both are possibly regulated by *Msc-1*. All of these genes were detected and confirmed by qRT-PCR. The presence of these genes suggests their possible role in tapetum and pollen exine formation in GMS pepper. Most key genes and transcription factors involved in these processes were down-regulated in the sterile line 16C1369A. This study provides a better understanding of GMS (*msc-1*) molecular functioning in pepper.

## 1. Introduction

The use of male sterile lines to produce hybrid seeds is one of the most important methods in hybrid breeding; it is used, for instance, in crops such as rice, maize, Chinese cabbage, sunflower and pepper [1,2] The genic male sterile (*msc-1*) (GMS) pepper is a spontaneous male-sterile mutation found in Shenjiao (*C. annuum*) in China [3]. To date, this GMS line has been used successfully to create several varieties of Shenjiao pepper; these are No. 3, 4, and 5 etc. widely grown in Northern China. 

The defects of stamens provide the biological basis for male sterility in plants; these include defects in both anther and pollen development. The anther consists of four distinct cell layers: epidermis, endothecium, middle layer, and tapetum. Microspores are produced in the locule surrounded by the tapetum. Pollen development relies heavily on the surrounding tissues that comprise the anther wall, especially the tapetum [4,5]. Abnormal tapetal development affects microspore development, eventually leading to male sterility [6].

The pollen wall, which consists of an outer layer (exine) and inner layer (intine) is encloses the pollen grain [7,8]. The pollen wall protects the sperm from harsh conditions, such as high temperatures, ultraviolet light, microbial attack, and water loss; the pollen wall also discriminates between own pollen grains and those of other individuals, to avoid inadequate fertilization that leads to embryo lethality [9]. It has been reported that most of the precursor materials of the pollen wall originate from the tapetum [10,11].

During anther development, the tapetum fulfils the role of sporopollenin biosynthesis for exine formation, synthesis of the callose (b-1,3-glucanase) complex that causes the release of microspores from the tetrad, and synthesis of numerous elaioplasts and cytoplasmic lipid bodies for pollen coat formation [12,13,14,15,16,17]. 

Pollen development from archesporial cells to mature pollen consists of a series of complex events regulated by a wide variety of genes [18,19]. In *Arabidopsis*, several genes regulate the tapetum and its functions. Among these genes, *DYSFUNCTIONAL TAPETUM1* (*DYT1*) and *ABORTED MICROSPORES* (*AMS*) encode a bHLH transcription factor [17,20]; Mutations in these two genes can cause excessive tapetal expansion into the locule and result in sporophytic male sterility [20,21]. *DEFECTIVE IN TAPETAL DEVELOPMENT AND FUNCTION1* (*TDF1*) and *Male Sterile 188* (*MS188*) encode a putative R2R3-MYB transcription factor [16,21], and *MALE STERILITY 1* (*MS1*) encodes a PHD transcription factor [22,23,24]. These genes form a genetic pathway (*DYT1-TDF1-AMS-MS188-MS1*) that regulates tapetum development and function [25]. 

Many loci and genes controlling male sterility have been reported in other horticultural crops. The male sterile (*ms10^35^*) mutant of cultivated tomato (*Solanum lycopersicum*) exhibits dysfunctional meiosis and an abnormal tapetum during anther development, resulting in no pollen production. And the tomato *Ms10^35^* has been identified as a master regulator controlling several genes involved in anther development [26]. A cucumber (*Cucumis sativus* L.) male sterile mutant (*ms-3)* caused defective microsporogenesis, resulting in no tetrad or microspores being formed, and *Csa3M006660* has been identified as the most likely candidate gene for *Ms-3* [27]. In cabbage (*Brassica oleracea* L.), *Bol0357N3* is thought to be the most likely candidate gene underlying the *ms-cd1* locus, which causes abortion of pollen development and male sterility [28]. In pepper, *Msc-1* encodes a homolog of AtDYT1, which has been found to control genic male sterility, and down-regulation of *Msc-1* can cause male sterility [29]. The *MS1*, a homolog of AtMS1, which encodes a PHD-type transcription factor that regulates pollen and tapetum development, has been identified as a strong candidate gene for the *ms1* gene [30]. The *CaAMS* gene is preferentially expressed in the tapetum at the tetrad and the early to mid uninucleate stages; down-regulation of *CaAMS* leads to abortive pollens in pepper flowers [31]. 

High-throughput technologies for measuring gene expression levels and protein abundance have enabled transcriptome- and proteome-level analyses of developmental processes, gene functions, adaptations, and physiological stress responses in plants [32] such as eggplant (*Solanum melongena* L.), pomegranate (*Punicagranatum* L.), rice (*Oryza sativa* L.), and tea (*Camellia sinensis* L.) [33,34,35,36]. Recently, proteomic approaches have been applied to the study of anther development in pepper. Wu et al. [37] analyzed the anther proteomes, comparing cytoplasmic male sterile (CMS) and maintainer lines; they revealed that DEPs are potentially associated with instability in metabolism, excessive ethylene synthesis, and starch synthesis reduction. Fang et al. [38] found that during pepper anther development, significant changes in amino acid synthesis occur in association with abnormal tapetum maturity; these amino acid changes are likely to be an important cause of male sterility in pepper. Guo et al. [39] used label-free mass spectrometry quantitation to compare protein expression profiles in a CMS pepper line and its maintainer line; they found that the differentially abundant proteins (DAPS) involved in pollen exine formation, pyruvate metabolic processes, the tricarboxylic acid cycle, the mitochondrial electron transport chain, and oxidative stress response might be related to pollen abortion in pepper. Liu et al. [40] analyzed the anther transcriptomes in pepper CMS lines and its fertility restoration; they found that anther transcriptome analysis may be useful for identifying potential candidate genes associated with the formation or abortion of pollen. It has been reported that the pollen abortion is associated with upregulation of the genes that code for methyltransferase during meiosis in CMS lines [41]. RNA sequencing has been used to identify many of the genes potentially involved in pollen development in fertile pepper [42]. 

Although we have previously identified a strong candidate gene for *msc-1* [29], we focus here on proteins, because they are the final products of genes and are more directly related to cellular metabolism and biological processes. Prior to now, the mechanisms by which these genes and proteins regulate male sterility in the GMS (*msc-1*) pepper have not been explained. In this study, we use proteomic and transcriptomic analysis to identify the DEPs and DEGs in GMS (*msc-1*) pepper and its male fertile line at the key stage of pollen abortion. We discuss the relationship between these DEPs and DEGs and male sterility in GMS pepper, examine the possible biological functions of these DEPs and DEGs, and explore their potential effects on anther development and male sterility.

## 2. Results

### 2.1. Cytological Observation of Pepper Anthers at Different Developmental Stages

To determine the cause of anther sterility in 16C1369A, pepper anthers of the sterile line 16C1369A and the fertile line 16C1369B were selected at four different pollen developmental stages and paraffin section analysis was performed. At the pollen mother cell stage, the sterile 16C1369A and fertile line 16C1369B did not differ significantly in their cytological structure (Figure 1A,D), whereas pollen abortion was observed in the sterile line 16C1369A in the tetrad stage (Figure 1B,E); in this stage, the sterile line 16C1369A tapetal layer cells were over-vacuolized and premature death occurred, whereas in the fertile line 16C1369B, the tapetal layer cells were degraded sufficiently for microspore development. In the sterile line, the 16C1369A tapetal layer cells subsequently tightly surrounded and squeezed the microspores, inhibiting callosum degradation and microspore release from the tetrad at the uninucleate stage (Figure 1F). After this, the microspores became dissociated and died during the later stages of pollen development. Finally, the collapsed tapetal cells and crushed microspores ultimately formed a dense belt (Figure 1H). Microspores were released normally from the tetrad in the fertile line 16C1369B, then became rounded, and developed a three-grooved germination pore; the tapetal layer cells dissolved, or partially remained on the endothecium (Figure 1C,G). Therefore, the key stage of pollen abortion occurred at the tetrad stage, due to premature tapetum death in the GMS pepper.

### 2.2. Overview of Quantitative Proteomics Analysis

In total, 502,216 spectra were generated using label-free analysis in the sterile line 16C1369A and fertile line 16C1369B. A total of 150 852 spectra matched to known spectra, and 29 056 peptides, 21 108 unique peptides, and 4909 protein species were identified (FDR ≤ 0.01) (Appendix A). The correlation coefficients (R^2^) between the biological replicates were 0.9845 (16C1369A-1 and 16C1369A-2), 0.9795 (16C1369A-1 and 16C1369A-3), 0.9734 (16C1369A-2 and 16C1369A-3), 0.9873 (16C1369B-1 and 16C1369B-2), 0.9895 (16C1369B-1 and 16C1369B-3), and 0.9885 (16C1369B-2 and 16C1369B-3), respectively. The distribution of molecular weights of the identified proteins, number of peptides, length of peptides and the coverage of the proteins by the identified peptides were provided in Appendix A. Based on the threshold for screening DEPs (|log_2_ FC| ≥ 1 and *p* ≤ 0.05), 674 DEPs were identified and quantified; 111 were up-regulated and 71 were down-regulated in the fertile line 16C1369B. Additionally, 276 and 216 protein species accumulated specifically in the sterile line 16C1369A and the fertile line 16C1369B, respectively (Figure 2A; Appendix A). The results indicate that these DEPs may form a complex regulatory network for pollen abortion in pepper.

### 2.3. Overview of the Transcriptomic Analysis

RNA-seq was performed using the anther from the sterile line 16C1369A and fertile line 16C1369B, and each line included three biological replicates. For each sample, 87–89% of the clean reads were mapped to the genome of pepper (http://peppersequence.genomics.cn/page/species/index.jsp), and 25,168, 24,941, 25,035, 25,454, 25,176, and 25,221 unique genes were detected in 16C1369A-1, 16C1369A-2, 16C1369A-3, 16C1369B-1, 16C1369B-2, and 16C1369B-3, respectively (Appendix A). The correlation coefficients (R^2^) between the biological replicates were 0.9995 (16C1369A-1 and 16C1369A-2), 0.9988 (16C1369A-1 and 16C1369A-3), 0.9991 (16C1369A-2 and 16C1369A-3), 0.9164 (16C1369B-1 and 16C1369B-2), 0.9816(16C1369B-1 and 16C1369B-3), and 0.8453 (16C1369B-2 and 16C1369B-3), respectively. DEGs were screened based on an absolute fold change value of |log_2_ FC| ≥ 1 and false discovery rate (FDR) ≤ 0.01. In total, 1069 DEGs were identified, among which 403 were down-regulated and 583 were up-regulated. Additionally, 15 and 68 genes accumulated specifically in the sterile line 16C1369A and fertile line 16C1369B, respectively (Figure 2B; Appendix A).

### 2.4. GO Annotation Function and KEGG Analysis of the DEPs and DEGs

Gene ontology (GO) analysis indicated that 182 DEPs were enriched in the following processes: “ribosome biogenesis” (biological process), “structural molecule activity” (molecular function), and “cytoplasmic part” (cellular component) (Appendix AA). There were 492 presence/absence proteins that were enriched in “pollen development” (biological process), “hydrolase activity, acting on glycosyl bonds” (molecular function) and “lysosome” (cellular component) (Appendix AB). We identified 1069 DEGs (651 up-regulated, 418 down-regulated), and these were enriched in the following processes: “DNA unwinding involved in DNA replication” (biological process), “DNA helicase activity” (molecular function) and “MCM complex” (cellular component) (Appendix AC).

We further performed KEGG pathway analysis on the identified DEPs and DEGs. The 182 DEPs were significantly enriched in the following categories: “ribosome” (18, 9.9%), “protein processing in endoplasmic reticulum” (13, 7.1%), “pyruvate metabolism” (7, 3.8%), “amino sugar and nucleotide sugar metabolism” (7, 3.8%), and “carbon metabolism” (7, 3.8%) (*p* ≤ 0.05; Appendix AA; Appendix A). And the 492 presence/absence proteins that were enriched in the following processes: “ubiquinone and other terpenoid-quinone biosynthesis” (7, 1.4%), “stilbenoid, diarylheptanoid, and gingerol biosynthesis” (4, 0.81%), “flavonoid biosynthesis” (4, 0.81%), and “cutin, suberine, and wax biosynthesis” (3, 0.61%) (Appendix AB; Appendix A). In terms of DEGs, the 20 most significantly enriched KEGG pathways are shown in Appendix AC; Appendix A. The detected DEGs were significantly enriched in these processes: “plant hormone signal transduction” (14, 8.0%), “phenylpropanoid biosynthesis” (13, 7.4%), “carbon metabolism” (11, 6.3%), “starch and sucrose metabolism” (10, 5.7%), and “glutathione metabolism” (10, 5.7%).

### 2.5. Comparison of Transcriptome and Proteome Data 

To evaluate the relationship between transcript levels and protein abundance, we compared all the transcripts and their corresponding quantified proteins in the sterile line 16C1369A and the fertile line 16C1369B at the tetrad stage. Among all the transcripts and the quantified proteins, there were 2478 genes that were found in both the transcriptome and proteome (Figure 2C). Of the 764 DEPs and 1069 DEGs identified, 52 cor-DEGs-DEPs were regulated at both the transcriptional and translational levels (Figure 2D).

We conducted a correlation analysis between the transcriptomic and quantitative proteomic data. The expression levels of all the transcripts and their corresponding quantified proteins in sterile line 16C1369A and fertile line 16C1369B showed limited correlation (*r* = 0.415 and 0.448; Figure 3A,B). However, less correlation was observed between all the transcripts and their corresponding quantified proteins (*r* = 0.242, Figure 3C). For the 764 DEPs and 1069 DEGs identified, the correlation of the 52 cor-DEGs-DEPs genes was 0.32 (Figure 3D), and a higher correlation was observed between proteins and their corresponding mRNAs with the same expression trend (*r* = 0.436; Figure 3E) or opposite expression trend (*r* = 0.773; Figure 3F).

### 2.6. Cluster Analysis of Expression Patterns in the cor-DEGs-DEPs Genes

Among the 52 cor-DEGs-DEPs genes, 40 genes showed the same expression trend, whereas 12 genes showed the opposite expression trend at the proteomic and transcriptomic levels (Figure 4; Appendix A). The correlated proteins and genes mostly showed up-regulated expression trends. We suggest that some of these genes might play important roles in pollen development.

### 2.7. GO and KEGG Analysis of the cor-DEGs-DEPs Genes 

Gene ontology (GO) analysis indicated that 45 of the 52 cor-DEGs-DEPs genes had GO annotations. The results covered a wide range of cellular components, molecular functions, and biological processes, including 30 important functional groups (Figure 5A). The largest subcategory found in the “biological process” was “external encapsulating structure organization”, followed by “pollen exine formation”, “pollen wall assembly”, “pollen development”, and others. For the “cellular component” category the three largest subcategories were “extracellular region”, “external encapsulating structure”, and “cell wall”. For the “molecular function” category, “carbon-nitrogen lyase activity”, “amine-lyase activity”, “strictosidine synthase activity”, and “fatty acid synthase activity” were the subcategories with the most annotations.

The KEGG pathways of the cor-DEGs-DEPs genes were identified using a *p*-value ≤ 0.05 as the cut-off, and the 52 cor-DEGs-DEPs genes were mapped to 47 KEGG pathways (Appendix A). The results showed that three KEGG pathways were significantly enriched at both mRNA and protein levels; these included the following pathways: “metabolic pathways” (map01100), “Biosynthesis of secondary metabolites” (map01110), and “Phenylpropanoid biosynthesis” (map00940) (Figure 5B). It has been reported that the “Phenylpropanoid biosynthesis” pathways are associated with pollen exine formation [43,44,45]. Therefore, based on the analysis of GO and the KEGG pathways enrichment, the difference between the sterile line 16C1369A and fertile line 16C1369B was mainly related to “pollen exine formation”, “pollen wall assembly”, “pollen development”, and “Phenylpropanoid biosynthesis”.

### 2.8. DEPs and DEGs Related to Anther and Pollen Development

Pollen development is a complex process that involves many events, especially tapetum development and pollen exine formation. In this study, based on the GO and KEGG analysis, we identified 13 DEPs and 14 DEGs involved the pollen development; most of these, including five cor-DEGs-DEPs genes (*TA29*, *TA1*, *FAS1*, *LAP3*, and *PG2*), were up-regulated in the fertile line 16C1369B. Among these genes, we propose that *AMS* and *TDF1* are involved in tapetum development. The genes *ACOS5*, *CYP704B1*, *PKSA,* and *PKSB*, *TKPR1*, *TKPR2*, *MS2,* and *ABCG26* were involved in the pollen exine formation (Table 1).

### 2.9. Validation of Gene Expression Levels 

qRT-PCR was performed to validate the RNA-seq results and the correspondence between protein levels and mRNA abundance. We selected 14 up- or down-regulated DEGs and 13 DEPs (including five cor-DEGs-DEPs genes) involved in tapetum and pollen development for qRT-PCR analysis. Twelve DEGs were highly expressed and two DEGs (*ARO1*, *AGL62*) were down-regulated in the fertile line 16C1369B. This result is consistent with the RNA-Seq results (Figure 6A). The genes involved in coding for the 13 DEPs were all expressed significantly more in the fertile line 16C1369B than in the sterile line 16C1369A, with the same trend in the label-free data (Figure 6B). These results indicate that the transcriptome and proteome data were reliable.

## 3. Discussion

Although genic male sterility provides great potential in terms of promoting pepper heterosis, no prior studies have characterized the *msc-1* changes in protein and gene expression in pepper anthers at the proteomic and the transcriptomic levels. As such, in order to gain a better understanding of the mechanisms of GMS regulation in pepper, we have conducted proteomic and transcriptomic analysis comparing sterile and fertile lines, in the tetrad stages of anther development.

In this study, the sterile line 16C1369A tapetal layer cells were over-vacuolized and premature death occurred (Figure 1E), and finally the collapsed tapetal cells and crushed microspores formed a dense belt (Figure 1H). In the sterile line, we observed that the tapetal layer cells were over-vacuolized and premature death occurred; this finding suggests abnormal processing of programmed cell death (PCD) in the sterile line, causing the tapetum to have defects in sporopollenin synthesis and nutrient supply, which affect later anther development. These results are consistent with the findings of Guo et al. [39] and Fang et al. [38] and suggest that the abnormal development of the tapetal cells and microspores leads to pollen and anther abortion. In addition, we have previously identified a strong candidate gene for *msc-1* [29]; Msc-1 is a homolog of AtDYT1, which is a bHLH transcription factor involved in early tapetal development [20].

The anther structure consists of gametophytes surrounded by four distinct cell layers: outer epidermis, endothecium, middle cell layer, and the innermost tapetum [46]. These layers, especially the tapetum is known to act as a supplier of metabolites, nutrients, and sporopollenin precursors [44]. Tapetum degeneration is induced through the process of PCD during the late developmental stage of the anther. Furthermore, the normal process of PCD in the tapetum is essential for normal exine pattern formation [47,48].

Exine consists primarily of sporopollenin, a tough and chemically inert biopolymer, which ensures the production of functional pollens grains [8]. Establishment of exine ornamentation by sporopollenin polymerization needs three key steps: synthesis, secretion, and translocation of sporopollenin precursors. Synthesis of a majority of the sporopollenin precursors is conducted in the tapetum [9]. Although the mechanisms of sporopollenin biosynthesis are not well understood, it appears that genes in pathways which produce fatty acids, phenylpropanoids, as well as some anther-specific genes, are important for exine synthesis [49,50,51]. 

In our proteomic data, many key proteins have been identified for pollen exine formation. These include ACOS5 (Capana02g003302), CYP704B1 (Capana01g002203), PKSA (Capana01g003460) and PKSB (Capana08g002676), TKPR1 (Capana05g000665), TKPR2 (Capana01g002831), MS2 (Capana03g003125) and ABCG26 (Capana07g002406), and these DEPs were all up-regulated in 16C1369B (Table 1) which was confirmed by the qRT-PCR analysis (Figure 6). In Arabidopsis, the proteins that we identified as being involved in pollen exine formation are reported to participate in the synthesis of sporopollenin precursors [9]. Among these, ACOS5 catalyzes mid-/long-chain fatty acids into fatty acyl-CoA in plastids; this is then hydroxylated by members of the cytochrome P450 superfamily, especially CYP703A2 and CYP704B1. The hydroxylated products are then catalyzed by PKSA and PKSB into triketide and tetraketide α-pyrones, which are the substrates of TKPR1and TKPR2, respectively [52,53,54]. MS2, a fatty acyl reductase, encodes a fatty acid reductase and catalyzes the palmitoyl acyl carrier protein into a fatty alcohol [55]. The resultant sporopollenin precursors are transported to the anther locule by a member of ATP-binding cassette transporter superfamily (ABCG26) and by lipid transfer proteins [56,57,58]. So due to the down-regulated of these proteins in the male sterile line 16C1369A, it may inhibit the synthesis and translocation of the sporopollenin precursors. 

In our transcriptomic data, two DEGs, *AMS* (*Capana08g000254*) and *TDF1* (*Capana04g001901*), were identified and were up-regulated in 16C1369B (Table 1; Figure 6). In Arabidopsis, both of these DEGs are essential for tapetum development [17,21,59]. In our previous study, we used genome resequencing to identify a strong candidate gene for *msc-1* [29]; there was a 7-bp deletion in the 3rd exon, which leads to a premature stop codon, and may cause a loss of function mutation in 16C1369A. In addition to the fact thatMsc-1 is a homology of DYT1, it was up-regulated in 16C1369B. In Arabidopsis, *DYT1* is the most upstream regulator in tapetum development, and directly binds the promoter region of *TDF1*; it thereby affects many downstream genes related to tapetum development and pollen wall formation [60]. *TDF1* directly regulates *AMS* via an AACCT cis-element [61,62]. *AMS* can bind to the promoters of *CYP703A2*, *PKSB*, *TKPR1*, and *CYP704B1* in vivo [59]. *MS188* interacts with *AMS* to regulate the expression of *CYP703A2* [63]. *MS188* directly regulates the expression of *PKSA*, *PKSB*, *MS2*, and *CYP703A2*; *MS188* and other MYB family members play redundant roles in activating the expression of *CYP704B1*, *ACOS5*, and *TKPR1* [54]. Therefore, in the sterile line 16C1369A, *AMS* (*Capana08g000254*) and *TDF1* (*Capana04g001901*) may be regulated by Msc-1, thus affecting tapetum development.

By comparing the transcriptome and proteome, we identified several cor-DEGs-DEPs involved in tapetum development and pollen exine formation. And all of these were significantly up-regulated in 16C1369B.

In this study, *Msc-1* was not listed as a DEP or a DEG in the proteome and the transcriptome, however *Msc-1* was detected both in the sterile line 16C1369A and fertile line 16C1369B with the log_2_ (16C1369B/16C1369A) was 0.558 in the transcriptome. And in our qRT-PCR analysis, *Msc-1* was highly expressed in the pollen mother cell stage in the fertile line 16C1369B and was decreased significantly with the anther development, besides that the expression level of *Msc-1* was much higher in the fertile line 16C1369B than in the sterile line 16C1369A in the four anther development stage (Figure 7A). So, based on our interpretations of our findings, we propose a model to explain the mechanisms of GMS in pepper (Figure 7B). Due to the down-regulation of *Msc-1* (*DYT1*) in the 16C1369A [29], down-regulation of *TDF1* and *AMS* may occur. This may result in abnormal PCD in the tapetum, severe inhibition of sporopollenin biosynthesis and transport, and defects in the formation of exine. Ultimately, this leads to pollen abortion in GMS pepper.

## 4. Materials and Methods 

### 4.1. Plant Materials

Genic male-sterile line 16C1369A and its near isogenic male-fertile line 16C1369B were grown in plastic-covered tunnels at the China Agricultural University, Beijing, China, in 2016. Flowers at different developmental stages were collected for paraffin section analysis, using Safranin O-Fast Green Staining. For proteomic analysis, RNA sequencing, and gene expression analysis, anthers were collected from flower buds at the key stage of pollen abortion, frozen in liquid nitrogen immediately, and then stored at −80 °C for later use. For proteomicand RNA-seq analysis, three biological replicates of sterile and fertile anthers (A1–A3; B1–B3) were collected from 150 to 200 flower buds at the tetrad stage, which were selected randomly from 30 pepper plants. For real-time PCR analysis, three biological and three technical replicates were conducted.

### 4.2. Protein Isolation, Enzymolysis, and Label-Free Analysis 

Anther protein extraction and label-free quantitative mass spectrometry analysis were performed according to Guo et al. [39]. For protein quantitation, we required at least two unique peptides in three biological replicates to be detected; the relative quantitative protein ratios between the two groups were calculated by comparing the average abundance values obtained in the three biological replicates. For analysis of differentially expressed proteins (DEPs), a 2-fold cutoff was used determine distinguish between up-regulated and down-regulated proteins. Additionally, proteins detected in only one of the groups (the sterile line 16C1369A or fertile line 16C1369B) at least two replicates were considered to be presence/absence proteins. To analyze differences between presence/absence protein species, the fold-change calculated was based on the mean value of the lowest protein intensity detected in each group.

### 4.3. RNA Isolation, Library Preparation, and Sequencing

Total RNA was extracted from anthers using TRIzol (Invitrogen, Waltham, MA, USA) according to the manufacturer’s instructions. Sequencing libraries were constructed using NEBNext^®^ Ultra™ RNA Library Prep Kit for Illumina^®^ (NEB, Ipswich, MA, USA) following manufacturer’s recommendations. The purity and quality of the libraries was assessed using an Agilent 2100 Bioanalyzer and Qubit 2.0. Then, the six libraries were sequenced using the Illumina Hiseq2500/X platform and 125/150 bp paired-end reads were generated. Clean data (clean reads) were obtained by removing reads containing adapter, reads containing ploy-N, and low-quality reads from raw data. At the same time, the Q20, Q30, and GC content of the clean data was calculated. All the downstream analyses were based on the clean data using high quality reads. TopHat2 was used to map clean reads to reference gene and reference genome [64]. Cufflinks were used to discover novel genes [65]. Cuffquant and Cuffnorm (v 2.2.1) were used to calculate Fragments Per Kilobase of transcript per Million mapped reads (FPKMs) of genes in each sample. Gene FPKMs were computed by summing the FPKMs of transcripts in each gene group. We used the DESeq2 R package [66] to analyze the differentially expressed genes (DEGs) in the sterile line 16C1369A and the fertile line 16C1369B.

### 4.4. Bioinformatics Analysis

Gene ontology (GO) enrichment analysis of DEGs and DEPs was implemented using the GOseq R package [67] in which gene length bias was corrected. Pathway analysis of DEGs and DEPs was performed using the Kyoto Encyclopedia of Genes and Genomes (KEGG) (http://www.genome.jp/kegg). A *p*-value ≤ 0.05 was used as the threshold to determine the significant enrichments of GO and KEGG pathways.

The RNA-Seq raw sequence data are deposited in the Short Read Archive (SRA) of National Centre for Biotechnology (NCBI) and are available under the accession: PRJNA524267. And the mass spectrometry proteomics data have been deposited to the ProteomeXchange Consortium via the PRIDE partner repository with the dataset identifier PXD012762. The reviewer account: Username: reviewer22878@ebi.ac.uk; Password: U964wSDm. 

### 4.5. Quantitative Real-Time PCR (qRT-PCR) Analysis

Total RNA of anthers from the sterile line 16C1369A and the fertile line 16C1369B was exacted using the SV Total RNA Isolation System Kit (Promega, Madison, WI, USA), following manufacturer’s instructions. We performed reverse transcription using 1 μg of total RNA using a PrimeScript™ RT Kit (Takara, Kusatsu, Shiga, Japan). Real-time PCR was carried out with using GoTaq^®^ qPCR Master Mix (Promega), following manufacturer’s instructions, on an ABI 7500 real-time PCR system. The thermocycling conditions were as follows: 95 °C for 2 min, 40 cycles of 95 °C for 15 s, and 60 °C for 1 min. Relative expression values were calculated using the 2^−ΔΔ*C*t^ method, and the *UBI-3*(AY486137.1) as the reference gene. The primers used for Real-time PCR were designed using Primer 5.0 (http://www.premierbiosoft.com/primerdesign/) and are listed in Appendix A. 

## 5. Conclusions

In this study, we identified numerous proteins and genes associated with pollen development and genic male sterility, using proteomic and transcriptomic analysis. In total, 764 DEPs and 1069 DEGs were identified at the proteomic and transcriptomic levels, respectively. Fifty-two cor-DEGs-DEPs were detected at both the proteomic and transcriptomic level. Bioinformatics analysis suggested that these DEPs and DEGs are involved in pollen exine formation, pollen wall assembly, pollen development, and phenylpropanoid biosynthesis. Based on our results, we propose a gene regulation network model to explain the mechanisms of genic male sterility in pepper (Figure 7). This study will improve our understanding of the genes associated with pollen development in the GMS pepper and contribute to the improvement of pepper hybrid breeding.

## Figures and Tables

**Figure 1 ijms-20-01789-f001:**
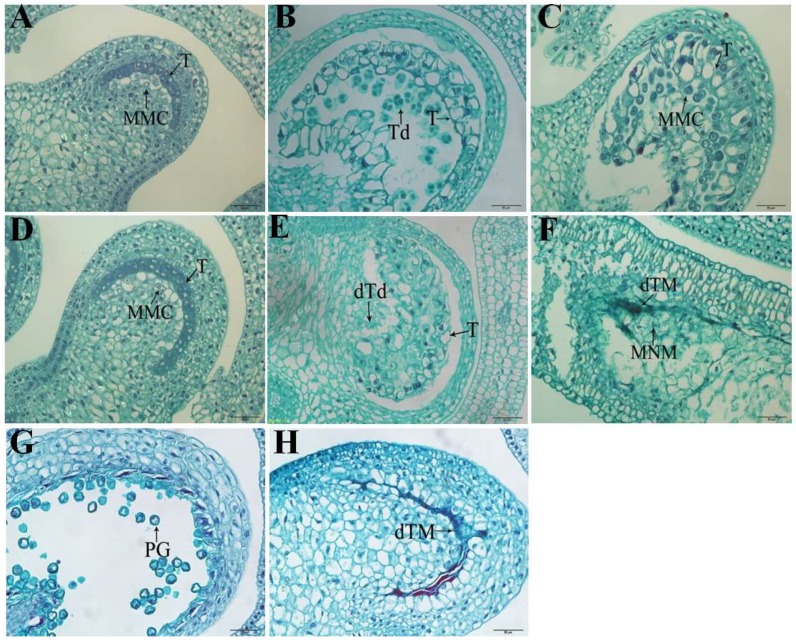
Pollen morphological features in anthers from the sterile line 16C1369A and fertile line 16C1369B. (**A**–**C**,**G**) Transverse sections of 16C1369B anthers; normal tapetum and mature anthers developed in the fertile line. (**D**–**F**,**H**) Transverse sections of sterile anthers; abnormal tapetum was formed, and pollen was not produced. (**A**,**D**) Pollen mother cell stage; (**B**,**E**) tetrad stage; (**C**,**F**) uninucleate stage; (**G**,**H**) mature stage. Td: tetrad; T: tapetum; MMC: microspore mother cell; dTd, death tetrad; MNM: mononuclear microspore; dTM: death tapetum and microspore; PG: pollen grain. Scale bars = 50 μm.

**Figure 2 ijms-20-01789-f002:**
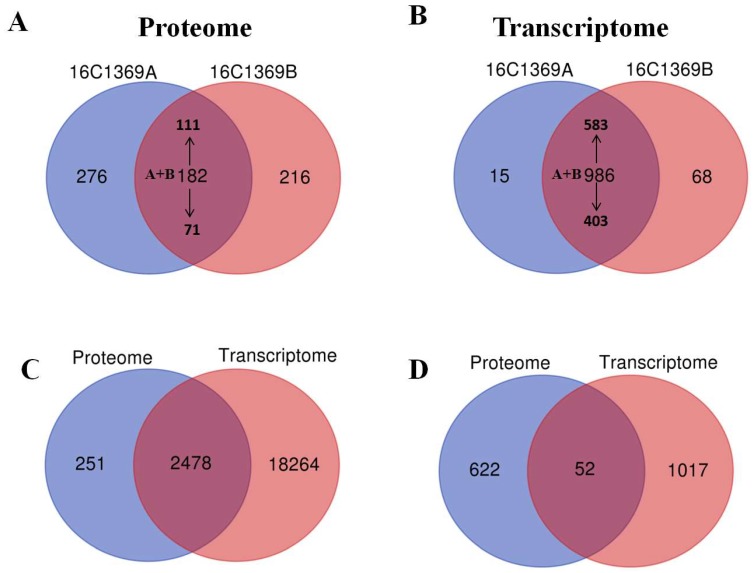
Venn diagram of transcriptome and proteome. (**A**) Comparison of differentially expressed proteins (DEPs) identified in the sterile 16C1369A and the fertile line 16C1369B; (**B**) comparison of differentially expressed genes (DEGs) identified in the sterile 16C1369A and the fertile line 16C1369B; (**C**) comparison of all the proteins and genes identified in the sterile 16C1369A and the fertile line 16C1369B; (**D**) comparison of all the DEPs and DEGs identified in the sterile 16C1369A and the fertile line 16C1369B. “↑” and “↓” indicate up-regulated and down-regulated, respectively. “A + B” indicates the DEPs and DEGs identified in both lines.

**Figure 3 ijms-20-01789-f003:**
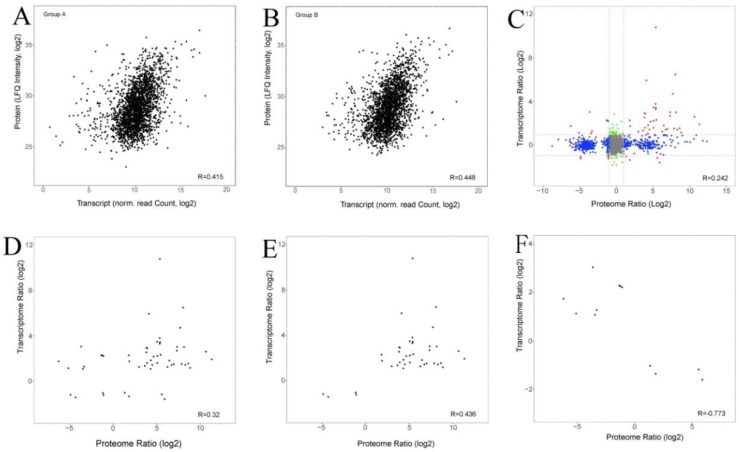
Comparison of expression ratios from transcriptome (y-axis) and proteome (x-axis) profiling. (**A**) Scatterplots of the relationship between genes quantified in both transcriptomic and proteomic analysis in 16C1369A; (**B**) Scatterplots of the relationship between genes quantified in both transcriptome and proteomic analysis in 16C1369B; (**C**) Scatterplots of the relationship between genes quantified in both transcriptomic and proteomic analysis; (**D**) Scatterplots and correlation coefficients between DEPs and DEGs in both transcriptomic and proteomic analysis; (**E**) Scatterplots and correlation coefficients between proteins and genes which showed the same types of changes in expression; (**F**) Scatterplots and correlation coefficients between proteins and genes which showed opposing changes in expression. Gray dots: absence of DEPs and DEGs; red dots: DEPs and DEGs; green dots: DEGs present and DEPs absent; blue dots: DEPs present and DEGs absent. The data were log2-transformed.

**Figure 4 ijms-20-01789-f004:**
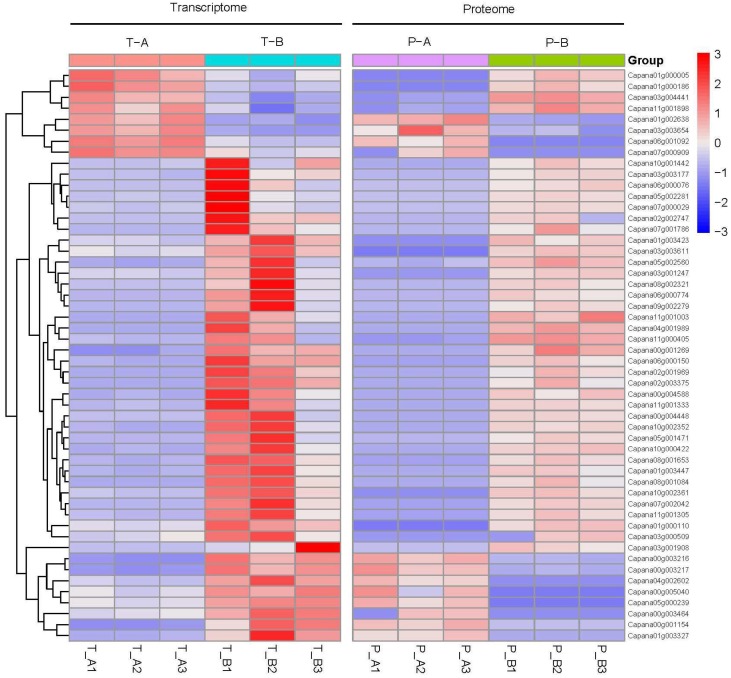
Comparison of changes in cor-DEGs-DEPs expression. T-A and T-B: transcriptomes of 16C1369A and 16C1369B, respectively. P-A and P-B: proteomes of 16C1369A and 16C1369B, respectively.The color code is as follows: Red indicates up-regulated cor-DEGs-DEPs genes; blue indicates down-regulated cor-DEGs-DEPs genes; white indicates unchanged cor-DEGs-DEPs genes. Each row represents the log2 (16C1369B/16C1369A) of a gene or protein. The color scale of the heat map ranges from saturated blue (value, −3.0) to saturated red (value, 3.0) in the natural logarithmic scale.

**Figure 5 ijms-20-01789-f005:**
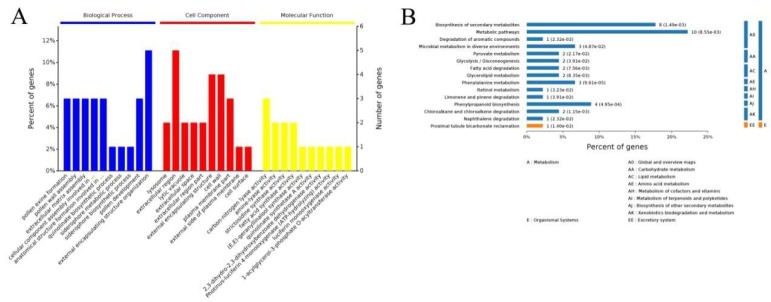
Gene ontology (GO) and Kyoto Encyclopedia of Genes and Genomes (KEGG) enrichment analysis of cor-DEGs-DEPs genes. (**A**) GO functional enrichment analysis of cor-DEGs-DEPs genes. (**B**) KEGG enrichment analysis of cor-DEGs-DEPs genes.

**Figure 6 ijms-20-01789-f006:**
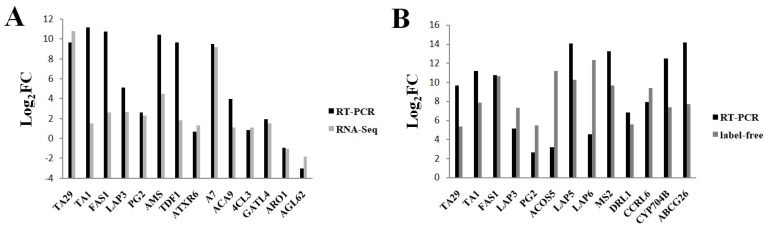
Validation of transcriptome and proteome data using qRT-PCR. (**A**) Relative mRNA abundance of genes selected from the DEGs; (**B**) Relative mRNA abundance of genes selected form the DEPs. Y-axis means the log_2_ (16C1369B/16C1369A) of a gene or protein.

**Figure 7 ijms-20-01789-f007:**
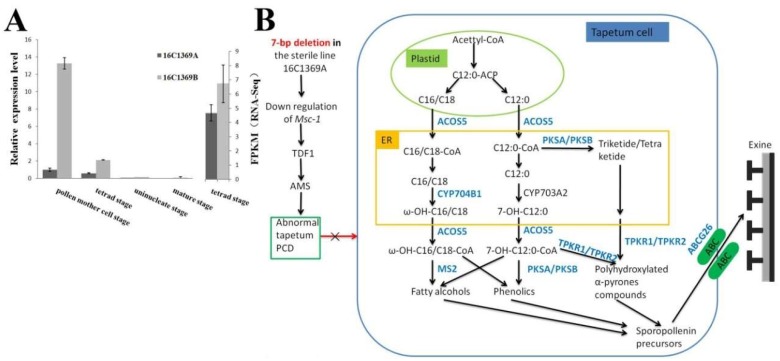
A possible gene regulation network in the male sterile line 16C1369A. (**A**) The expression pattern of *Msc-1* detected in qRT-PCR and transcriptome. (**B**) A possible gene regulation network according to which abnormal tapetal programmed cell death (PCD) is triggered by the down-regulation of *Msc-1* in the male sterile line 16C1369A.

**Table 1 ijms-20-01789-t001:** List of DEPs and DEGs associated with anther and pollen development in pepper.

**DEPs**	**Annotation**	**Gene ID**	**Description**	**Homologous**	**Regulation (B/A)**
	ACOS5	Capana02g003302	ACYL-COA SYNTHETASE 5	AT1G62940	UP
	PKSA	Capana01g003460	Chalcone and stilbene synthase family protein	AT1G02050	UP
	PKSB	Capana08g002676	Chalcone and stilbene synthase family protein	AT4G34850	UP
	MS2	Capana03g003125	Jojoba acyl CoA reductase-related male sterility protein	AT3G11980	UP
	TKPR1	Capana05g000665	Dihydroflavonol 4-reductase-like 1	AT4G35420	UP
	TKPR2	Capana01g002831	NAD(P)-binding Rossmann-fold superfamily protein	AT1G68540	UP
	CYP704B	Capana01g002203	Cytochrome P450, family 704, subfamily B	AT1G69500	UP
	ABCG26	Capana07g002406	ATP-binding cassette transporter G26	AT3G13220	UP
**DEGs**	**Annotation**	**Gene ID**	**Description**	**Homologous**	**Regulation**
	AMS	Capana08g000254	ABORTED MICROSPORES	AT2G16910	UP
	TDF1	Capana04g001901	DEFECTIVE IN MERISTEM DEVELOPMENT AND FUNCTION 1	AT3G28470	UP
	ATXR6	Capana03g001971	ARABIDOPSIS TRITHORAX-RELATED PROTEIN 6	AT5G24330	UP
	A7	Capana07g001721	ANTHER 7	AT4G28395	UP
	ACA9	Capana03g000026	AUTOINHIBITED CA(2+)-ATPASE 9	AT3G21180	UP
	4CL3	Capana03g001733	4-COUMARATE:COA LIGASE 3	AT1G65060	UP
	GATL4	Capana01g003063	GALACTURONOSYLTRANSFERASE-LIKE 4	AT3G06260	UP
	ARO1	Capana08g002699	ARMADILLO REPEAT ONLY 1	AT4G34940	DOWN
	AGL62	Trans_newGene_14655	AGAMOUS-LIKE 62	AT5G60440	DOWN
**cor-DEGs-DEPs**	**Annotation**	**Gene ID**	**Description**	**Homologous**	**Regulation**
	FAS1	Capana06g000774	Fasciclin-like arabinogalactan family protein	AT5G16920	UP
	LAP3	Capana03g003177	Calcium-dependent phosphotriesterase superfamily protein	AT3G59530	UP
	PG2	Capana11g001305	polygalacturonase 2	AT1G70370	UP
	TA29	Capana02g001969	TA29_TOBAC Anther-specific protein TA-29	NA	UP
	TA1	Capana04g001989	Arabidopsis homolog of TASSELSEED2. Expressed specifically in tapetal cells	AT3G42960	UP

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
