# Peer review of "Complementary Transcriptomic and Proteomic Analysis Reveals a Complex Network Regulating Pollen Abortion in GMS (msc-1) Pepper (Capsicum annuum L.)"

_ijms, 2019, doi:10.3390/ijms20071789_

Round 1
Reviewer 1 Report
This is a very interesting research aimed to investigate key genes involved in male sterily in pepper.
There are some issue which need to be clarified:
1) Related to qRT-PCR the author did not report reference genes in supplementary table 7 as well in the text. I think they have to indicate which refence genes have been used and the condition of amplification
2) From a breeding point of view it could be of interest within this work to identify specific molecular markers for the selection of ms line. i did not find any results about this and no mention in the discussion. Could the authors give some explanation of this lack
3) I advice to avoid acronyms in the abstract, just put the entire name and then acronym in parentheses
Other minor issues:
L18 space repeated after the dot
L24 check space repeated
L38 .....is a spontaneous male‑sterile mutation found in Shenjiao...could the author provide a reference?
L38 C annuum var.??
L158 Paragraph 2.3 Instead to give all the numbers which make difficult the reading, just put the table S3 in the main text this will give to the reader a better view. In the paragraph, the authors should briefly sintetize the results
Author Response
Dear Editors and Reviewer 1,
Thank you for paying attention to our manuscript (Manuscript ID: ijms-463499). We also thank the reviewer for his (or hers) helpful suggestions. In this version, we have studied comments carefully and have made correction which we hope meet with approval. The main corrections in the manuscript and the responds to the reviewer’s comments are as follow:
Point 1: Related to qRT-PCR the author did not report reference genes in supplementary table 7 as well in the text. I think they have to indicate which reference genes have been used and the condition of amplification
Response 1: In this study we used the“UBI-3(AY486137.1)” as the reference gene, and it was listed in the supplementary table 7. And meanwhile we have added the related information the manuscript in to the M&M part (line: 423-424).
Point 2: From a breeding point of view it could be of interest within this work to identify specific molecular markers for the selection of ms line. i did not find any results about this and no mention in the discussion. Could the authors give some explanation of this lack
Response 2: In our previous work we had identified a strong candidate gene for msc-1, and developed a gene marker (indel -12) for the selection of this ms line (Cheng et al. 2018). So in this study we did not discussed the molecular markers for the msc-1 line.
Reference: Cheng, Q.; Wang, P.; Liu, J.; Wu, L.; Zhang, Z.; Li, T.; Gao, W.; Yang, W.; Sun, L.; Shen, H. Identification of candidate genes underlying genic male-sterile msc-1 locus via genome resequencing in Capsicum annuum L. Theor Appl Genet. 2018, 131(9), 1861-1872.
Point 3: I advice to avoid acronyms in the abstract, just put the entire name and then acronym in parentheses
Response 3: We have used the entire name instead of the acronyms in the abstract in this revision.
Other minor issues:
Point 4:: L18 space repeated after the dot
Response 4: We have deleted a space after the dot (line: 18).
Point 5: L24 check space repeated
Response 5: We have deleted a space after the “(AMS)” (line: 25).
Point 6: L38 .....is a spontaneous male‑sterile mutation found in Shenjiao...could the author provide a reference?
L38 C annuum var.??
Response 6: We have added a reference in this revised manuscript (line: 40), and the reference were “Wang, D., Bosland, P W. The genes of Capsicum. HortSci. 2006, 41,1169-1187”.
As “L38 C annuum var.??” we did not know the reviewer’ meaning. Does this description is wrong?
Point 7: L158 Paragraph 2.3 Instead to give all the numbers which make difficult the reading, just put the table S3 in the main text this will give to the reader a better view. In the paragraph, the authors should briefly sintetize the results
Response 7: The table S3 was the overall results of the RNA sequencing and assembly, which just as the table S1 was the overall results of the protein identification. And in this study we emphasized the DEGs, so we put it in the supplementary materials.
Reviewer 2 Report
In this manuscript by Cheng et al., the authors perform transcriptome and proteome profiling of male sterile line 16C1369A and the corresponding fertile line 16C1369B. While the datasets presented could be potentially valuable, I have several concerns about the data and conclusions. Throughout the manuscript the authors cite their prior work identifying the gene msc-1 as the causal mutant for the male sterile phenotype and that this gene is down-regulated in the mutant. However, this gene is not listed as a DEG or DEP in this study. This is concerning since the group’s prior publication noted that this gene is expressed and DE specifically in the anther. The reason for this discrepancy needs to be fully explained if the datasets presented are to be trusted.
Beyond the confusion with dataset quality, data analysis could be improved. Most notably, adjusted p-values need to be used to call DEPs, and p-values need to be reported for all lines in Table S2. Additionally, adjusted p-values need to be calculated for GO and KEGG analyses and presented in a readable way (the text in the supplemental figures is too small and low resolution). The KEGG analysis in Figure 5B needs to be larger in order to be readable and should only include significant terms. The methods used for figures need to be more fully explained in all figure legends – for example, I am not able to fully evaluate Figure 4 due to the lack of clarity on what values are being plotted.
Comments on figures:
Figure 1: The figure legend needs additional details about the developmental stages shown in each image.
Figures 3 and 5: Axis labels need to be much larger to be readable.
Figure 4: The color scale in the heat maps are not described, and it is unclear what the values plotted correspond to (expression? Fold-change? Neither of these seems likely given the data). This information needs to be presented in the figure and the legend.
Figure 6: Figure legend needs to state how fold-change was calculated for this plot.
Figures S1 – S3: Figure labels need to be larger and higher resolution.
Table S2: p-values need to be listed for all DEPs.
Figure 7: It is not clear where the evidence for this model came from.
Author Response
Dear Editors and Reviewer 2,
Thank you for paying attention to our manuscript (Manuscript ID: ijms-463499). We also thank the reviewer for his (or hers) helpful suggestions. In this version, we have studied comments carefully and have made correction which we hope meet with approval. The main corrections in the manuscript and the responds to the reviewer’s comments can be loaded below.

Round 2
Reviewer 1 Report
The authors have improved the manuscript and answered properly to all comments.
At L38 replace "C. annuum var." with "C. annuum"
Author Response
Dear Editors and Reviewer 1,
Thank you for paying attention to our manuscript (Manuscript ID: ijms-463499). We also thank the reviewer for his (or hers) helpful suggestions. In this version, we have studied comments carefully and have made correction which we hope meet with approval. The main corrections in the manuscript and the responds to the reviewer’s comments are below:
Point 1: At L38 replace "C. annuum var." with "C. annuum"
Response 1:We have replaced “C. annuum var.” with “C. annuum” (line 40).
Reviewer 2 Report
While several minor concerns have been updated in this revision, my two major concerns were not fully addressed. Additional modifications to the text would be needed to address these concerns.
Point 1: Expression change of Msc-1 in this transcriptome dataset should be presented in a main figure even if the log2FC did not meet the expression cutoff. All of the conclusions of this manuscript depend on this gene showing differential expression, so that data must be presented.
Point 2: While FDR values were added to supplemental tables, they were not used to filter significant terms. In the methods, you added that FDR < 0.01 was your cutoff for the KEGG analysis, however terms above this value were included in Figure 5. In fact, the only significant term is phenylalanine metabolism. The results need to be updated and re-written with your modified cutoffs.
Line 249 should read p-value < 0.05.
Author Response
Dear Editors and Reviewer 2,
Thank you for paying attention to our manuscript (Manuscript ID: ijms-463499). We also thank the reviewer for his (or hers) helpful suggestions. In this version, we have studied comments carefully and have made correction which we hope meet with approval. The main corrections in the manuscript and the responds to the reviewer’s comments are below:
Point 1: Expression change of Msc-1 in this transcriptome dataset should be presented in a main figure even if the log2FC did not meet the expression cutoff. All of the conclusions of this manuscript depend on this gene showing differential expression, so that data must be presented.
Response 1:We have presented the expression pattern of Msc-1 in transcriptome and in different anther development in the main figure (Figure 7A). And revised the related information in the main text (line: 356-362).
Point 2: While FDR values were added to supplemental tables, they were not used to filter significant terms. In the methods, you added that FDR < 0.01 was your cutoff for the KEGG analysis, however terms above this value were included in Figure 5. In fact, the only significant term is phenylalanine metabolism. The results need to be updated and re-written with your modified cutoffs.
Response 2:In this study, the FDR values was used to filter the DEPs and DEGs, and we used the p-value ≤ 0.05 to filter significant terms. We added the FDR < 0.01 in the KEGG and GO analysis was our mistake. And we are very sorry for that mistake. We have amended the related information (line: 418). The KEGG results were described based on the number of genes or proteins which term were enriched. So we do not modify the KEGG results.
Point 3: Line 249 should read p-value < 0.05.
Response 3:We have modified the p-value (line: 249).
Round 3
Reviewer 2 Report
Thank you for incorporating the requested changes.